# Mean Velocity of the Pulmonary Artery as a Clinically Relevant Prognostic Indicator in Patients with Heart Failure with Preserved Ejection Fraction

**DOI:** 10.3390/jcm11030491

**Published:** 2022-01-19

**Authors:** Blanca Trejo-Velasco, Ignacio Cruz-González, Manuel Barreiro-Pérez, Elena Díaz-Peláez, Pilar García-González, Ana Martín-García, Rocío Eiros, Soraya Merchán-Gómez, Candelas Pérez del Villar, Oscar Fabregat-Andrés, Francisco Ridocci-Soriano, Pedro L. Sánchez

**Affiliations:** 1Cardiology Department, University Hospital of Salamanca, Institute of Biomedical Research of Salamanca (IBSAL), 37007 Salamanca, Spain; cruzgonzalez.ignacio@gmail.com (I.C.-G.); manuelbarreiroperez@gmail.com (M.B.-P.); elenadp1985@gmail.com (E.D.-P.); anamartin.amg@gmail.com (A.M.-G.); eirosbachiller@gmail.com (R.E.); arayamergo@hotmail.com (S.M.-G.); cperezdelvillar@usal.es (C.P.d.V.); pedrolsanchez@me.com (P.L.S.); 2Biomedical Research Networking Center on Cardiovascular Diseases (CIBERCV), 28029 Madrid, Spain; 3Cardiology Department, University Hospital Alvaro Cunqueiro, 36213 Vigo, Spain; 4Cardiac Magnetic Resonance Imaging Unit, ASCIRES, 46015 Valencia, Spain; pilugarciagonzalez@hotmail.com; 5Cardiology Department, IMED Hospital of Valencia, 46100 Burjassot, Spain; osfabregat@gmail.com; 6Cardiology Department, General University Hospital of Valencia, 46014 Valencia, Spain; fridoccs@me.com; 7Department of Medicine, University of Valencia, 46010 Valencia, Spain

**Keywords:** heart failure preserved ejection fraction (HFpEF), pulmonary hypertension, right ventricle, right heart unit coupling, outcomes

## Abstract

Background: Right ventricular (RV) to pulmonary circulation (PC) coupling can stratify prognosis in heart failure (HF). In this study, we assessed the prognostic role of the mean velocity of the pulmonary artery (mvPA) determined by cardiac magnetic resonance (CMR) in HF with preserved ejection fraction (HFpEF). Methods: Inclusion of 58 HFpEF outpatients that underwent CMR with measurement of RV–PC coupling parameters including mvPA between 2016 and 2019. The primary combined endpoint was a composite of HF readmissions and all-cause mortality. Results: Optimal cut-off value of mvPA calculated by receiver operating curve for the prediction of the primary endpoint was 9 cm/s. Over a median follow-up of 23 months (interquartile range: 24), 21 patients met the primary endpoint. The primary endpoint was more frequent in patients with mvPA ≤ 9 cm/s, as indicated by Kaplan–Meier survival curves; Log-Rank: 9.193, *p* = 0.02, regardless of RV dysfunction. On Cox multivariate analysis, mvPA ≤ 9 cm/s emerged as an independent prognostic predictor of the primary endpoint (HR: 4.11, 95% CI: 1.28–13.19, *p* = 0.017), together with left atrial area by CMR (HR: 1.08, 95% CI: 1.01–1.24, *p* = 0.034). Conclusions: In our HFpEF cohort, mvPA was associated with a higher rate of the primary endpoint, regardless of RV function, thus enabling identification of patients at higher risk of cardiovascular events before structural damage onset.

## 1. Introduction

Heart failure with preserved ejection fraction (HFpEF) is a prevalent condition that entails a high morbidity and mortality burden [1,2]. Among HFpEF patients, right ventricular (RV) dysfunction and pulmonary hypertension (PH) are common associated conditions that convey a worse prognosis [3,4,5,6]. Over the past years, the assessment of RV contractile function relative to its load, i.e., RV to pulmonary circulation (PC) coupling, has arisen as a tool to improve prognostic stratification in HF, as it allows for an earlier identification of patients at increased risk of adverse events than either RV dysfunction or PH separately [7,8].

The gold standard measure of RV–PC coupling is the ratio of RV end-systolic elastance to effective arterial elastance (Ees/Ea), but this parameter is rarely assessed in clinical practice, as it requires invasive measurements by means of right heart catheterization (RHC) as well as specific, dedicated material. Instead, most studies have evaluated the prognostic role of non-invasive surrogates of the coupling status [9,10,11].

Recently, several monocentric studies have reported on the potential value of the mean velocity of the pulmonary artery (mvPA) determined by cardiac magnetic resonance (CMR) as a novel non-invasive surrogate of the RV–PC coupling unit, with prognostic value in HF with reduced (HFrEF) and mid-range ejection fraction (HFmEF) [12,13]. Unlike other indexes of RV–PC coupling, the mvPA is not directly calculated from RV stroke volume nor RV ejection fraction (RVEF) and could allow for further prognostic stratification both in patients with and without associated RV dysfunction. Indeed, lower mvPA values have been associated with a higher risk of subsequent HF admissions and death before onset of structural RV damage. However, the value of mvPA in HFpEF has not been studied to date. In this study, we aimed to determine if mvPA displays a similar prognostic role in HFpEF as it does in patients with HFrEF.

## 2. Materials and Methods

### 2.1. Study Population

This retrospective cohort study was conducted in a single tertiary care teaching hospital between January 2016 and January 2019. In total, 112 consecutive patients with confirmed HFpEF diagnosis according to clinical practice guidelines [1] that underwent CMR assessment with measurement of RV–PC coupling parameters during their initial diagnostic work-up at the outpatient’s clinic were included. Overall, 18 patients with severe valvular heart disease, 15 unable to undergo CMR on account of advanced kidney disease or claustrophobia, 9 with insufficient follow-up data, and 12 with no evidence of left ventricular (LV) diastolic dysfunction were excluded, leaving a total study sample of 58 patients. Medical therapy was optimized according to guidelines. The study was conducted according to the guidelines of the Declaration of Helsinki and approved by the local Ethics Committee of the University Hospital of Salamanca, and informed consent was obtained from all subjects involved in the study.

### 2.2. Transthoracic Echocardiography

A comprehensive transthoracic echocardiography (TTE) was performed in all patients. Echocardiography measurements were recorded and averaged over three consecutive heart cycles in patients in sinus rhythm and over 3–5 heart cycles in patients in atrial fibrillation (AF). Diastolic function was assessed according to published guidelines [1,14]. In addition, a thorough study of the right heart was performed, including determination of tricuspid annular plane systolic excursion (TAPSE) in M-mode, quantification of the degree of tricuspid regurgitation (TR) [15], and estimation of systolic pulmonary artery pressure (SPAP) using the peak velocity of the TR jet derived from continuous-wave Doppler and the RV–PC coupling indicator TAPSE/SPAP ratio [7].

### 2.3. Cardiac Magnetic Resonance

CMR was performed with 1.5 T CMR equipment (Philips Healthcare, Best, The Netherlands) in stable, euvolemic patients [16]. In patients with AF, the ventricular response rate was controlled prior to performing CMR. Standard ECG gated breath-hold balanced steady-state free precession (bSSFP) cine sequences were employed for cine imaging. Long- and short-axis slices were acquired in order to evaluate ventricular volumes and function while the ejection fraction was calculated based on short-axis slices, according to Simpson’s method. A standard 17-segmented cardiac model was used for segmentation. Dimeglumine gadobenate 0.5 M contrast was injected intravenously for the assessment of late gadolinium enhancement (LGE), which was assessed on inversion-recovery bSSFP sequences. RV dysfunction was defined as RVEF ≤ 45% in agreement with prior studies evaluating RV dysfunction by CMR [17,18].

Pulmonary artery (PA) flow was assessed on slices perpendicular to the main PA, employing velocity-encoded gradient echo sequences. The axial section of the main PA was contoured in each cardiac phase to determine the PA area including minimum and maximum PA areas, as well as PA flow during the complete cardiac cycle. A dedicated software (Intellispace Portal 7.0, Philips Healthcare, Best, The Netherlands) calculated mvPA as the integral of the velocity in each of the voxels included within the PA outline over the complete cardiac cycle, Figure 1A–C. PA pulsatility was determined as [(maximum PA area − minimum PA area)/minimum PA area × 100], and pulmonary vascular resistance (PVR) was calculated by means of the formula: {PVR in Wood Units (WU) = 19.38 − [4.62 × Ln mvPA (cm/s)] − [0.08 × RV ejection fraction (RVEF)(%)]}, previously validated in PH patients [19,20]. Finally, RV to PC coupling ratio, which is the ratio between RV end-systolic maximal elastance (Emax, index of contractility) divided by PA effective elastance (Ea, index of arterial load), was estimated with the equation: [Emax/Ea = stroke volume(SV)/end-systolic volume(ESV)], validated by a prior study as an indicator of the RV–PC coupling state [21]. SV and ESV values in this equation were obtained by CMR.

### 2.4. Invasive Pressure Assessment

RHC was performed in 28 (48.3%) patients, in which this technique was clinically indicated, at the discretion of the patients’ physician. The procedure was conducted in the outpatient setting in stable, euvolemic patients, employing standard fluoroscopy guided Seldinger technique, through the basilic or femoral veins. Right chambers’ and pulmonary pressures were recorded at end-expiration in the supine position. Cardiac output was determined either by Fick or thermodilution methods, as appropriate. PVR, pulse pressure, transpulmonary gradient (TPG), and PA compliance were calculated employing standard formulas.

### 2.5. Clinical Follow-Up

The primary combined endpoint was defined as the composite of HF readmissions and all-cause death during follow-up. Data collection was performed through a centralized electronic health record system. 

### 2.6. Statistical Analysis

Continuous variables were expressed as mean ± standard deviation or median (interquartile range (IQR)), as appropriate, while discrete variables were expressed as percentages. MvPA was evaluated as a continuous variable and then categorized according to its optimal threshold to predict the primary combined endpoint at follow-up. This value was calculated by means of a receiver operating characteristic (ROC) sensitivity/1-specificity curve, as the value attaining a largest area under the curve (AUC). Patients were divided in two groups according to mvPA values. The Shapiro–Wilk test was employed to assess whether variables in both groups followed a normal distribution or not. Comparisons between both groups were made by χ^2^ test and unpaired Student’s *t*-test or Mann–Whitney–Wilcoxon test, as appropriate. The association between mvPA and the primary combined endpoint during follow-up was evaluated by Kaplan–Meier by means of the log-rank test. In addition, a multivariate Cox regression analysis was performed, including all variables with a *p*-value ≤ 0.10 on univariate analysis. The prognostic performance of mvPA was compared to that of other established parameters evaluating RV–PC coupling by means of ROC curve analysis and Cox multivariable regression analysis. A 2-sided *p*-value ≤ 0.05 was considered statistically significant. SPSS for Windows (v.21.0 Statistical Package for the Social Sciences, International Business Machines, Inc., Armonk, New York, NY, USA) was employed for statistical analysis.

## 3. Results

Baseline characteristics of the 58 included patients are summarized in Table 1 and Table 2. Mean age was 67.5 ± 13.5, and 58.6% patients were male. Arterial hypertension was the most prevalent comorbidity in 51.7% subjects, followed by AF in 46.6%. One-fifth of patients had been previously admitted for decompensated HF, and over one-quarter maintained a NYHA functional class III–IV/IV on follow-up. On CMR, mean LV ejection fraction was 59.5 ± 8.7%, and 12 (20.7%) patients exhibited RV dysfunction. PH estimated by TTE as SPAP > 35 mmHg was present in 69% of patients, while RHC confirmed PH in 24 (41.4%) cases, accounting for 85.7% of patients subjected to this examination, Appendix A. Medical therapy at last follow-up included betablockers in 29 (50%) patients, ACE-II inhibitors or angiotensin-2 receptor blockers in 32 (55.2%), mineralocorticoid antagonists in 18 (31%), and diuretics in 50 (86.2%).

Median follow-up was 23 months (IQR 24 months). During this period, 21 patients met the primary endpoint on account of 15 hospital admissions for decompensated HF and 8 all-cause deaths. Death was preceded by a HF admission in four cases.

More patients in an advanced NYHA functional class suffered cardiovascular adverse events on follow-up, as compared to those on functional class I–II/IV, Table 1. On CMR, larger PA and left atrial areas, higher PVR, and lower mvPA values and LGE were associated with an increased number of events during follow-up (Table 2). No significant differences among other clinical, imaging, or hemodynamic parameters assessed by RHC existed between patients that developed the primary combined endpoint and those who remained event-free during follow-up. 

### 3.1. Baseline Characteristics According to mvPA

The optimal cut-off value of mvPA calculated by the ROC curve for the prediction of the primary endpoint was 9 cm/s, (AUC:0.75 (0.62–0.88), *p* = 0.002), Figure 2. This threshold was coincident with the median value of mvPA in our sample (9 cm/s; IQR 5.2).

Patients with mvPA ≤ 9 cm/s presented a higher number of cardiovascular events during follow-up at the expense of an increased rate of HF admissions, without differences in all-cause mortality, Table 3.

No significant differences in clinical or analytical baseline characteristics according to mvPA values were observed, Appendix A. 

Patients with mvPA ≤ 9 cm/s displayed greater LV hypertrophy and longer transmitral E-wave deceleration times. No significant differences in other echocardiographic parameters including RV function assessed by TAPSE, SPAP, or the TAPSE/SPAP ratio existed between patients with mvPA above and below 9 cm/s (Table 3). Notwithstanding, RVEF assessed by CMR was lower among patients with mvPA ≤ 9 cm/s. Patients with mvPA ≤ 9 cm/s also displayed enlarged RV end-systolic volumes, higher PVR and Ea/Emax estimated by CMR, and greater maximal and minimal PA areas, although no differences in PA pulsatility were observed. Regarding RHC parameters, there were no substantial differences in mean PA pressure, TPG, PVR, or PA compliance according to mvPA values, Table 3, Figure 1D.

### 3.2. Prognostic Performance of mvPA and Non-Invasive RV–PC Coupling Parameters

On univariate analysis, mvPA was associated with the primary combined endpoint (hazard ratio (HR): 1.36, 95% confidence interval (CI): 1.10–1.68, *p* = 0.004). The primary combined endpoint occurred more frequently among patients with reduced mvPA values ≤ 9 cm/s, as indicated by Kaplan–Meier analysis; log-rank: 9.193, *p* = 0.02, Figure 3A. Of importance, mvPA maintained its prognostic value irrespective of underlying RV dysfunction, Figure 3B,C, log-rank 8.905, *p* = 0.003.

The ability of mvPA to predict the primary combined endpoint during follow-up was compared to that of other established non-invasive parameters that also assess RV to PC coupling, i.e., TAPSE/SPAP and Ea/Emax ratios and to left atrial area on CMR, which displayed prognostic significance in our sample.

Unlike mvPA, TAPSE/SPAP ratio as a continuous variable was not associated with the primary combined endpoint (HR: 0.271, (95% CI: 0.02–3.86), *p* = 0.335) and exhibited a lower prognostic performance than mvPA on ROC curve analysis (AUC 0.620 (0.449–0.791), *p* = 0.164), Figure 4A. However, when analyzed according to its median value (0.43), the TAPSE/SPAP ratio displayed a significant association with the primary combined endpoint both on univariate (HR: 0.28, (95% CI: 0.08–0.96), *p* = 0.043) and Kaplan–Meier survival analysis, log-rank: 4.948, *p* = 0.026, Figure 4B. On the other hand, the Ea/Emax ratio was not associated with the primary combined endpoint, neither as a continuous variable nor after categorization according to its median value (0.76). Left atrial area was associated with a higher incidence of cardiovascular adverse events on univariate analysis (HR: 1.25 (95% CI: 1.07–1.46), *p* = 0.004) but exhibited a lower sensitivity than mvPA for the detection of the primary combined endpoint, despite a similar AUC value (AUC: 0.740, (95% CI: 0.59–0.90), *p* = 0.004), Figure 4A.

In order to determine the prognostic value of mvPA as an independent prognostic indicator in our sample, a multivariate Cox proportional hazard analysis was performed, including all variables that displayed a *p*-value below 0.10 on univariate analysis.

To avoid colinearity, TAPSE/SPAP, Ea/Emax ratios, maximal and minimal PA areas as well as PVR estimated by CMR, which integrates mvPA into its formulae, were evaluated on Cox multivariable regression against mvPA. As only mvPA maintained its prognostic significance, the other variables were discarded from the final model. 

On Cox regression analysis, only mvPA ≤ 9 cm/s (HR: 4.11, 95% CI: 1.28–13.19, *p* = 0.017) and left atrial area by CMR (HR: 1.12, 95% CI: 1.01–1.24, *p* = 0.034) remained as statistically significant predictors of the primary combined endpoint, Table 4.

## 4. Discussion

In this single-center study in HFpEF patients, mvPA was associated with an increased rate of the primary combined endpoint encompassing HF readmissions and all-cause death, alongside with left atrial area assessed by CMR. Of note, the prognostic value of mvPA was maintained both in patients with and without associated RV dysfunction, thus allowing for early risk stratification in HFpEF prior to onset of substantial structural damage.

Previous studies have reported on the value of mvPA in HFrEF and HFmEF [12,13]. However, to the best of our knowledge, this is the first one to ascertain its prognostic role in patients with HFpEF. This finding is relevant, as HFpEF determines a high burden of hospital admissions and is a major cause of death. Moreover, mvPA may contribute to improve the phenotypical classification of the heterogeneous HFpEF syndrome, which could help identify specific targeted therapies that improve prognosis in certain HFpEF phenotypes [2].

PH is a frequent HFpEF phenotype that can develop in up to 80% of HFpEF patients [5] and is a major determinant of RV dysfunction, which appears in as much as 30–50% of HFpEF patients [4,18]. In HF, heightened left atrial and pulmonary capillary pressures (PCP) increase RV net afterload at the expense of a higher pulsatile relative to resistive RV load, leading to impairment of the RV reserve, even in the absence of pulmonary vascular remodeling [8,22,23,24,25]. Indeed, abnormal RV to PC coupling develops in early phases in HF, before PVR rises, as a result of reduced PA compliance and increased PA stiffness [23]. Accordingly, an integral assessment of RV function relative to its load, i.e., RV to PC coupling, has recently emerged as a strong prognostic predictor in HF as well as in PH [7,8,9,10,11,12,26,27].

In clinical practice, RV–PC is generally assessed by means of non-invasive surrogate indicators. The TAPSE/SPAP ratio has been most frequently studied given its simplicity and the broad availability of TTE-derived measurements [7,9,10,11,28]. However, the intrinsic limitations of the echocardiographic parameters that compose this index might lessen its prognostic performance. On the other hand, non-invasive Ea/Emax estimation by CMR assumes negligible PCP values, which are not taken into account in its calculation, and is thus less reliable in HF as it is in patients with precapillary PH [21,26,27].

In our sample, mvPA displayed a greater prognostic performance than both indexes of RV–PC coupling and than PA maximal and minimal areas, which have also been linked with higher PA pressures and a worse prognosis in a variety of conditions besides HF [29]. Left atrial area by CMR was also prognostically relevant in our sample. Notwithstanding, we believe that mvPA is more useful as a tool to identify HF patients at a higher risk of adverse events, as its higher sensitivity reduces the risk of missing individuals with a poorer prognosis that would benefit from a more thorough clinical surveillance, while its lower specificity does not imply the use of any unnecessary treatments or examinations. We speculate that the fact that left atrial size is also influenced by of other factors such as atrial fibrillation might reduce its accuracy at reflecting a status of chronic congestion. In addition, the superior prognostic capacity of mvPA could be related to the fact that it combines information on RV function and RV afterload in a single parameter. On the one hand, mvPA reflects RV function and stroke volume, which generate power to pump blood forwards into the PA and thus generate fluid velocity at this level. In addition, mvPA also incorporates the interaction of forward RV flow with the PA vasculature, which depends not only on pulmonary vascular remodeling but also on PA buffering function, closely related to PA compliance. In consequence, mvPA constitutes a good non-invasive indicator of the RV–PA unit coupling state, as it describes the energy transfer between the RV and the pulmonary vasculature, encompassing both ventricular contractility and arterial afterload.

Of note, patients with mvPA ≤ 9 cm/s presented features of more advanced diastolic dysfunction and lower RVEF values as patients with mvPA > 9 cm/s, but baseline clinical characteristics did not differ significantly. These data reinforce the notion that increased LV filling pressures are associated with reduced compliance of the pulmonary vasculature and, subsequently, lower RV reserve and decreased RV-PC coupling. Importantly, mvPA ≤ 9 cm/s predicted worse outcomes not only across the whole sample but also enabled further prognostic stratification in patients with and without associated RV dysfunction (Figure 3). Indeed, patients with mvPA ≤ 9 cm/s and preserved RV function presented a higher incidence of cardiovascular adverse events during follow-up than patients with RV dysfunction but mvPA > 9 cm/s. Accordingly, mvPA can be useful to identify patients with HFpEF at higher risk of subsequent cardiovascular events before RV dysfunction onset, which will ultimately lead to end-stage circulatory failure.

Altogether, mvPA arises as a non-invasive, simple, non-operator-dependent parameter for the assessment of RV–PC coupling in HFpEF patients, which can improve the prognostic assessment of this population. Notwithstanding, larger studies confirming the prognostic role of mvPA are imperative in order to establish firm conclusions regarding the performance of this index in HFpEF.

### Limitations

The main limitations of this study stem from its retrospective design and the losses of some patients to follow-up, as mentioned in the Methods section. Additionally, we lacked a central core lab to assess imaging data, although both echocardiography and CMR were performed by experienced cardiologists with specific training on cardiac imaging. Moreover, RHC was only performed in a small subset of the total sample at the discretion of the patient’s physician given the invasive nature of this technique, so that a stronger association between CMR- and RHC-derived parameters cannot be discarded. Further studies including more patients undergoing RHC should evaluate the prognostic value of mvPA measured by CMR in HFpEF, with and without associated PH. Additionally, it is mandatory that our results be confirmed in different populations before mvPA is adopted into clinical practice. Finally, optimal HF guideline directed medical therapy did not include sodium-glucose cotransporter-2 (SGLT-2) inhibitors, which lacked evidence for the management of HFpEF at the time the study was performed.

## 5. Conclusions

In our single-center cohort of HFpEF patients, mvPA estimated by CMR was associated with an increased risk of the primary combined endpoint encompassing HF readmissions and death. Reduced mvPA values allowed for further prognostic stratification both in patients with and without associated RV dysfunction. Of importance, mvPA outperformed the prognostic value of previously established non-invasive RV–PC coupling indicators such as the TAPSE/SPAP ratio.

## Figures and Tables

**Figure 1 jcm-11-00491-f001:**
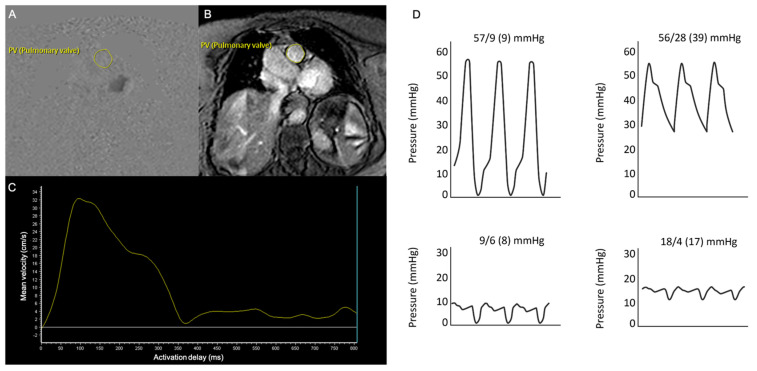
Cardiac magnetic resonance and right heart catheterization examinations of a representative patient. (**A**,**B**) Velocity-encoded gradient echo sequences on an axial section of the main PA. (**C**) Offline analysis of PA flow rate vs. time to calculate average and peak PA velocities. (**D**) Right chamber’s pressures determined by right heart catheterization.

**Figure 2 jcm-11-00491-f002:**
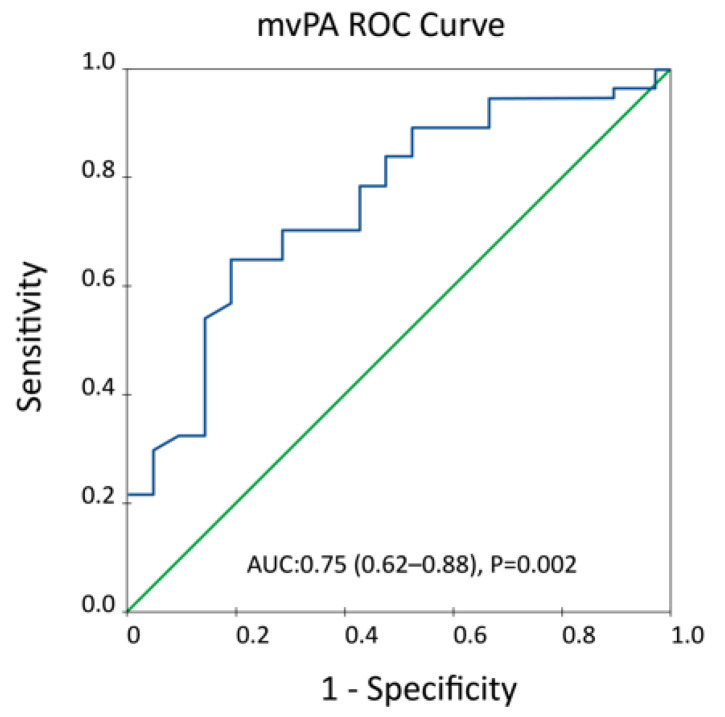
Estimation of mean velocity pulmonary artery (mvPA) optimal threshold according to ROC sensitivity/1-specificity curve to predict the primary combined endpoint.

**Figure 3 jcm-11-00491-f003:**
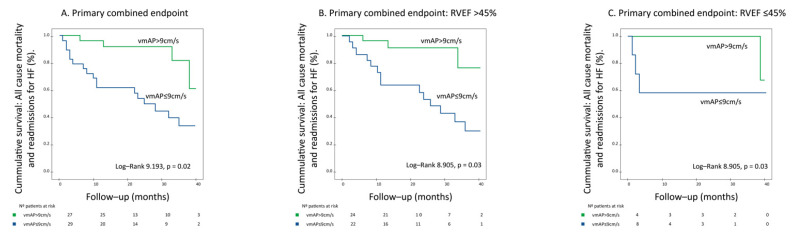
Survival analysis according to mvPA, as shown by Kaplan–Meier curves. (**A**) Reduced mvPA ≤ 9 cm/s was associated with higher rates of the primary combined endpoint. (**B**,**C**) The prognostic value of mvPA ≤ 9 cm/s for the prediction of the primary combined endpoint was maintained in patients with and without associated RV dysfunction.

**Figure 4 jcm-11-00491-f004:**
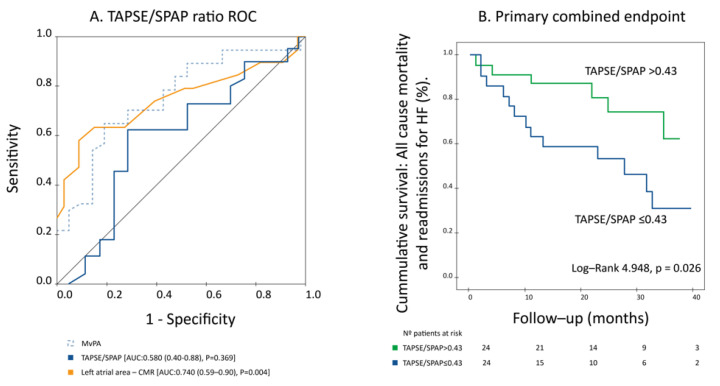
(**A**) Left atrial area displayed lower sensitivity than mvPA for the detection of the primary combined endpoint, despite a similar AUC value, while TAPSE/SPAP ratio had a much lower performance for the prediction of the primary combined endpoint, which did not attain statistical significance. (**B**) Notwithstanding, a TAPSE/SPAP ratio ≤ 0.43 was associated with higher rates of the primary combined endpoint.

**Table 1 jcm-11-00491-t001:** Baseline clinical characteristics according to the primary combined endpoint.

	No Event (*n* = 37)	Primary Combined Endpoint (*n* = 21)	Total Sample (*n* = 58)	*p*-Value
Age (years)	67.2 ± 12.7	68 ± 15.2	67.5 ± 13.5	0.391
Sex, male (*n*, %)	20 (54.1)	14 (66.7)	34 (58.6)	0.349
BSA (m^2^)	1.8 ± 0.16	1.8 ± 0.18	1.8 ± 0.17	0.512
Arterial hypertension (*n*, %)	17 (45.9)	13 (61.9)	30 (51.7)	0.242
Diabetes mellitus (*n*, %)	7 (18.9)	3 (14.3)	10 (17.2)	0.653
Dislipidemia (*n*, %)	16 (43.2)	8 (38.1)	24 (41.4)	0.702
Atrial fibrillation (*n*, %)	17 (45.9)	10 (47.6)	27 (46.6)	0.902
LBBB	2 (5.4)	3 (14.3)	5 (8.6)	0.247
Previous coronary artery disease (*n*, %)	5 (13.5)	3 (14.3)	8 (13.8)	0.935
Glomerular filtration rate (mL/min/1.73 m^2^)	72.3 ± 17.6	73.9 ± 18.5	72.9 ± 17.8	0.691
Stage 3–4 chronic kidney disease (*n*, %)	5 (13.5)	1 (4.8)	6 (10.3)	0.293
NT-proBNP (pg/mL)	841.3 ± 343.5	1341.1 ± 474.2	1281.8 ± 1164.5	0.310
Prior HF hospitalization (*n*, %)	7 (18.9)	5 (23.8)	12 (20.7)	0.659
NYHA functional class (*n*, %)				
I	17 (45.9)	2 (9.5)	19 (32.8)	0.002
II	14 (37.8)	10 (47.6)	24 (41.4)	
III	6 (16.2)	4 (19)	10 (17.2)	
IV	0	5 (23.8)	5 (8.6)	
NYHA III–IV/IV (*n*, %)	6 (16.2)	9 (42.9)	15 (25.9)	0.026
Cerebrovascular disease (*n*, %)	5 (13.5)	3 (14.3)	8 (13.8)	0.935

BSA: body surface area. HF: heart failure. LBBB: left bundle branch block. Nt-proBNP = N-terminal brain natriuretic type peptide. NYHA = New York Heart Association.

**Table 2 jcm-11-00491-t002:** Baseline imaging and hemodynamic parameters according to the primary combined endpoint.

	No Event (*n* = 37)	Primary Combined Endpoint (*n* = 21)	Total Sample (*n* = 58)	*p*-Value
Echocardiography parameters				
LVEF (%)	58.9 ± 9.5	57.2 ± 8.4	58.3 ± 9.1	0.914
LV septum width (mm)	12.2 ± 1.7	12.8 ± 1.8	12.4 ± 1.7	0.267
LV posterior wall width (mm)	10.9 ± 2	11.5 ± 1.7	11.1 ± 1.9	0.244
LVEDD (mm)	47.5 ± 7.1	46.9 ± 5.9	47.2 ± 6.6	0.739
LVESD (mm)	33.7 ± 11.1	30.7 ± 9.7	32.5 ± 10.5	0.423
Indexed left atrial volume (mL/m^2^)	48.8 ± 18.7	43.9 ± 18.1	47.1 ± 18.5	0.405
E/A ratio	1.2 ± 0.7	1.0 ± 0.6	1.2 ± 0.7	0.440
DT (ms)	206.3 ± 55.1	212.2 ± 34.9	208.4 ± 48.1	0.773
e’ (septal)	6.6 ± 1.6	6.3 ± 2.5	6.5 ± 1.9	0.357
e’ (lateral)	10.4 ± 3.2	9.8 ± 4.2	10.2 ± 3.5	0.715
E/e’ ratio (lateral)	7.7 ± 3.5	7.1 ± 3.1	7.5 ± 3.3	0.631
TAPSE (mm)	21.5 ± 4.1	19.8 ± 5.3	20.9 ± 4.6	0.123
S’ tricuspid (cm/s)	11.3 ± 2.2	10.3 ± 4.1	10.9 ± 3	0.064
Pulmonary acceleration time (ms)	87.5 ± 26.3	83.6 ± 20.9	85.9 ± 24	0.646
PAPs (mmHg)	45.0 ± 16.6	46.6 ± 15.7	45.6 ± 16.1	0.532
TAPSE/PAPs	0.53 ± 0.2	0.46 ± 0.2	0.5 ± 0.2	0.369
TR grade ≥ 3/4	7 (18.9)	4 (19)	11 (19)	0.990
CMR parameters				
LVEF (%)	60.7 ± 8.1	57.4 ± 9.4	59.5 ± 8.7	0.165
iLVEDV (mL/m^2^)	82.2 ± 23.2	79.2 ± 34.2	81.1 ± 27.4	0.481
iLVESV (mL/m^2^)	34.5 ± 18.8	36.5 ± 22.8	35.2 ± 20.1	0.752
Left ventricular mass (g)	71.9 ± 21.1	72.1 ± 25.5	72 ± 22.5	0.984
RVEF (%)	55.5 ± 11.7	52.5 ± 9.1	54.4 ± 10.9	0.120
iRVEDV (mL/m^2^)	94.3 ± 24.7	106.3 ± 39.9	98.5 ± 31.1	0.233
iRVESV (mL/m^2^)	42.1 ± 15.9	52 ± 26.8	45.6 ± 20.7	0.160
LGE (*n*, %)	10 (27)	13 (61.9)	23 (39.7)	0.009
LGE ischemic pattern (*n*, %)	3 (8.1)	2 (9.5)	5 (8.6)	0.854
LGE non-ischemic pattern (*n*, %)	8 (21.6)	12 (57.1)	20 (34.5)	0.006
Left atrial area (mm^2^)	15.4 ± 3.4	19.6 ± 5.5	16.9 ± 4.7	0.005
Right atrial area (mm^2^)	15.7 ± 4.9	18.7 ± 10.4	16.8 ± 7.5	0.252
Maximal PA area (cm^2^)	8.4 ± 2.5	11.4 ± 3.4	9.5 ± 3.2	<0.001
Minimal PA area (cm^2^)	6.7 ± 2.0	9.2 ± 2.8	7.6 ± 2.6	<0.001
PA pulsatility (%)	26.9 ± 14.2	25.8 ± 19.3	26.7 ± 16.1	0.382
Right ventricular Ea/Emax	0.91 ± 0.58	0.96 ± 0.38	0.93 ± 0.51	0.120
mvPA (cm/s)	10.9 ± 3.9	7.7 ± 2.7	9.8 ± 3.9	0.001
PVR-CMR (Wood Units)	4.2 ± 2.3	5.9 ± 1.8	4.8 ± 2.3	0.001
RHC parameters *				
Mean PA pressure (mmHg)	33.6 ± 15.6	35.7 ± 14.7	34.5 ± 14.9	0.728
Pulmonary capillary pressure (mmHg)	15.9 ± 4.7	14.8 ± 5.4	15.5 ± 4.5	0.347
PA pulse pressure (mmHg)	34.3 ± 16.2	31.2 ± 10.5	32.9 ± 13.9	0.873
Cardiac index (mL/min/m^2^)	2.5 ± 0.5	2.5 ± 1.1	2.5 ± 0.8	0.506
PVR-CCD (UW)	4.4 ± 3.1	5.2 ± 2.9	4.8 ± 3.0	0.494
Transpulmonary gradient (mmHg)	17.7 ± 13.7	20.8 ± 14.7	19.1 ± 13.9	0.478
PA compliance (mL/mmHg)	2.3 ± 1.7	1.8 ± 0.5	2.1 ± 1.9	0.882

* Values available for *n* = 28 patients. CMR = cardiac magnetic resonance. DT = deceleration time. Ea = effective elastance. Emax = right ventricular maximal end-systolic elastance. LGE = late gadolinium enhancement. LV = left ventricular. LVEF = left ventricular ejection fraction. LVEDD = left ventricular end-diastolic diameter. LVESD = left ventricular end-systolic diameter. LVEDV = left ventricular end-diastolic volume. LVESV = left ventricular end-systolic volume. MvPA = mean velocity at the pulmonary artery. PA = pulmonary artery. PVR = pulmonary vascular resistance. TAPSE = tricuspid annular plane excursion.

**Table 3 jcm-11-00491-t003:** Baseline imaging and invasive hemodynamic parameters according to mvPA.

	mvAP ≤ 9 cm/s (*n* = 30)	mvAP > 9 cm/s (*n* = 28)	Total Sample (*n* = 58)	*p*-Value
Echocardiography parameters				
LVEF (%)	59.1 ± 10.2	57.3 ± 7.7	58.3 ± 9.1	0.460
LV septum width (mm)	12.9 ± 1.8	11.7 ± 1.4	12.4 ± 1.7	0.010
LV posterior wall width (mm)	11.8 ± 1.8	10.3 ± 1.8	11.1 ± 1.9	0.011
LVEDD (mm)	47.5 ± 6.9	46.9 ± 6.3	47.2 ± 6.6	0.784
LVESD (mm)	31.8 ± 11.3	33.5 ± 9.4	32.5 ± 10.5	0.648
Indexed left atrial volume (mL/m^2^)	44.9 ± 18.6	49.9 ± 18.3	47.1 ± 18.5	0.362
E/A ratio	1.1 ± 0.5	1.4 ± 0.8	1.2 ± 0.7	0.154
DT (ms)	224 ± 38.7	185 ± 53.2	208.4 ± 48.1	0.044
e’ (septal)	5.9 ± 2.2	7.1 ± 1.5	6.5 ± 1.9	0.069
e’ (lateral)	9.1 ± 3.9	11.1 ± 2.9	10.2 ± 3.5	0.059
E/e’ ratio (lateral)	7.3 ± 3.4	7.7 ± 3.3	7.5 ± 3.3	0.732
TAPSE (mm)	20.8 ± 5	21.1 ± 4.1	20.9 ± 4.6	0.674
S’ tricuspid (cm/s)	11 ± 3.6	10.7 ± 1.8	10.9 ± 3	0.730
Pulmonary acceleration time (ms)	82.9 ± 28.1	90.8 ± 14.9	85.9 ± 24	0.358
PAPs (mmHg)	46.5 ± 17.2	44.6 ± 15.1	45.6 ± 16.1	0.673
TAPSE/PAPs	0.49 ± 0.3	0.50 ± 0.2	0.5 ± 0.2	0.974
TR grade ≥ 3/4	7 (23.3)	4 (14.3)	11 (19)	0.380
CMR parameters				
LVEF (%)	58.7 ± 9.5	60.4 ± 7.7	59.5 ± 8.7	0.732
iLVEDV (mL/m^2^)	82 ± 30.9	80.1 ± 23.6	81.1 ± 27.4	0.779
iLVESV (mL/m^2^)	36.9 ± 24.2	33.4 ± 14.8	35.2 ± 20.1	0.932
Left ventricular mass (g)	73.7 ± 21.9	70.2 ± 23.5	72 ± 22.5	0.486
RVEF (%)	51 ± 11.4	57.9 ± 9.3	54.4 ± 10.9	0.015
iRVEDV (mL/m^2^)	106 ± 30.1	90.8 ± 30.7	98.5 ± 31.1	0.053
iRVESV (mL/m^2^)	51.3 ± 16.6	39.7 ± 23	45.6 ± 20.7	0.002
LGE (*n*, %)	14 (46.7)	9 (32.1)	23 (39.7)	0.215
LGE ischemic pattern (*n*, %)	2 (6.7)	3 (10.7)	5 (8.6)	0.610
LGE non-ischemic pattern (*n*, %)	13 (43.3)	7 (25)	20 (34.5)	0.117
Left atrial area (mm^2^)	18 ± 5.1	15.8 ± 4.1	16.9 ± 4.7	0.090
Right atrial area (mm^2^)	18.4 ± 9.4	15.2 ± 4.6	16.8 ± 7.5	0.128
Maximal PA area (cm^2^)	10.9 ± 3	7.8 ± 2.5	9.5 ± 3.2	<0.001
Minimal PA area (cm^2^)	8.8 ± 2.4	6.2 ± 2.2	7.6 ± 2.6	<0.001
PA pulsatility (%)	24.1 ± 14.1	29.3 ± 17.8	26.7 ± 16.1	0.194
Right ventricular Ea/Emax	1.1 ± 0.6	0.8 ± 0.4	0.93 ± 0.51	0.008
mvPA (cm/s)	6.8 ± 1.6	12.9 ± 3	9.8 ± 3.9	<0.001
PVR-CMR (Wood Units)	6.5 ± 1.8	3.1 ± 1.1	4.8 ± 2.3	<0.001
RHC parameters *				
Mean PA pressure (mmHg)	34.9 ± 14.8	33.6 ± 16.2	34.5 ± 14.9	0.823
Pulmonary capillary pressure (mmHg)	15.9 ± 5.5	14.6 ± 3.7	15.5 ± 4.5	0.513
PA pulse pressure (mmHg)	32.2 ± 13.6	34.4 ± 15.1	32.9 ± 13.9	0.698
Cardiac index (mL/min/m^2^)	2.5 ± 0.9	2.5 ± 0.4	2.5 ± 0.8	0.932
PVR-CCD (UW)	4.6 ± 2.8	5.2 ± 3.7	4.8 ± 3.0	0.804
Transpulmonary gradient (mmHg)	19.1 ± 13.5	19 ± 15.8	19.1 ± 13.9	0.993
PA compliance (mL/mmHg)	2.1 ± 1.3	2 ± 1.4	2.1 ± 1.9	0.904
Cardiovascular events				
Readmission for decompensated heart failure (*n*, %)	11 (36.7)	4 (14.3)	15 (25.9)	0.049
All-cause death (*n*, %)	6 (20)	2 (7.1)	8 (13.8)	0.156
Primary combined endpoint (*n*, %)	17 (56.7)	4 (14.3)	21 (36.2)	0.001

* Values available for *n* = 28 patients. CMR = cardiac magnetic resonance. DT = deceleration time. Ea = effective elastance. Emax = right ventricular maximal end-systolic elastance. LGE = late gadolinium enhancement. LV = left ventricular. LVEF = left ventricular ejection fraction. LVEDD = left ventricular end-diastolic diameter. LVESD = left ventricular end-systolic diameter. LVEDV = left ventricular end-diastolic volume. LVESV = left ventricular end-systolic volume. MvPA = mean velocity at the pulmonary artery. PA = pulmonary artery. PVR = pulmonary vascular resistance. TAPSE = tricuspid annular plane excursion.

**Table 4 jcm-11-00491-t004:** Multivariate regression analysis.

	Hazard Ratio (95% CI)	*p*-Value
NYHA functional class III–IV/IV	0.67 (0.24–1.90)	0.453
Left atrial area—CMR (mL/m^2^)	1.12 (1.01–1.24)	0.034
Late gadolinium enhancement—CMR	0.72 (0.24–2.12)	0.546
mvPA < 9 cm/s	4.11 (1.28–13.19)	0.017

CMR: cardiac magnetic resonance. NYHA: New York heart association. mvPA: mean velocity pulmonary area.

## Data Availability

The data presented in this study are contained within the article and Appendix A. Further data are available on request from the corresponding author.

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
