# Peer review of "Mean Velocity of the Pulmonary Artery as a Clinically Relevant Prognostic Indicator in Patients with Heart Failure with Preserved Ejection Fraction"

_jcm, 2022, doi:10.3390/jcm11030491_

Round 1

Reviewer 1 Report

The study sought to assess the value of mvAP, a CMR criteria, surrogate for right arterio-ventricular coupling, to improve stratification of patients with HFPEF.

The study is well conducted but many points need to be underlined. The most important one is the assertion mvPA is a strong prognostic predictor in patients with HFPEF. It’s a large overestimation of the result of the study. At best mvPA < 9m/s is associated with worse prognosis in a little, monocentric cohort. Authors state mvPA cut-off found in this study is in agreement with the one found in studies performed in other HF population. Nevertheless, both studies cited as references (ref 12 and 13) were performed by the same team. As a consequence, agreement is not surprising neither is the “coincidence” with of the cut-off with the median value of the population (see for example https://journals.plos.org/plosone/article?id=10.1371/journal.pone.0081699) On this basis, this cut-off cannot be described as a prognostic marker but only  as "associated with a different outcome in this specific population". This point must be underline in the discussion, as well as the absolutely needed confirmatory study in a different population.

The assertion of the additive value over RV dysfunction seems largely overestimated too. Why the authors used a RVEF 45% cut-off to perform a subgroup analysis? It’s always a bad idea to dichotomize a continuous variable, there are no explanation about the choice of this particular cut-off, and the subgroup of patient with RVEF 45% is by far too small to be relevant by itself or comparing to the RVEF<45% group

Moreover, looking at the ROC curve, sensibility and specificity don’t look awesome. Can the authors specify them and discuss the risk of misdiagnosis and misclassification?

As left atrial volume is a predictor of the primary endpoint, can the authors analyze the ROC curve of left atrial volume and show mvPA is effectively a better predictor than the widely used left atrial volume. Can the authors precise the indicators included in their univariate and multivariate analysis mentioned line 246-248. Why mvPA was not included as a continuous variable in this analysis?

Right Heart Catheterization was performed in a subset of patients. As it is a retrospective study and this test was performed on the only basis of physicians’ judgment, there is unavoidable bias. Demographic, CMR and echocardiographic data should be displayed (at least as a part of the supplemental data) to compare patients in whom RHC was done and those in whom it wasn’t.

In the “limitations” section, authors stated about the lost of follow-up for some patients. Are they writing about the nine patients mentioned line 73 or are there other losses?

In the introduction (line 55 and subsequent), it would be more accurate to state the authors’ team studied mvPA as a potential measurement of interest on monocentric small cohorts rather than presenting it as an established measurement ; or  please provide other references from other team

Author Response

Response to reviewers.

REVIEWER 1:

The study sought to assess the value of mvAP, a CMR criteria, surrogate for right arterio-ventricular coupling, to improve stratification of patients with HFPEF.

The study is well conducted but many points need to be underlined.

> The most important one is the assertion mvPA is a strong prognostic predictor in patients with HFPEF. It’s a large overestimation of the result of the study. At best mvPA < 9m/s is associated with worse prognosis in a little, monocentric cohort. Authors state mvPA cut-off found in this study is in agreement with the one found in studies performed in other HF population. Nevertheless, both studies cited as references (ref 12 and 13) were performed by the same team. As a consequence, agreement is not surprising neither is the “coincidence” with of the cut-off with the median value of the population (see for example https://journals.plos.org/plosone/article?id=10.1371/journal.pone.0081699) On this basis, this cut-off cannot be described as a prognostic marker but only  as "associated with a different outcome in this specific population". This point must be underline in the discussion, as well as the absolutely needed confirmatory study in a different population.

- Thank you for your important comment. We have now modified several points of the results and discussion sections according to this observation.

> The assertion of the additive value over RV dysfunction seems largely overestimated too. Why the authors used a RVEF 45% cut-off to perform a subgroup analysis? It’s always a bad idea to dichotomize a continuous variable, there are no explanation about the choice of this particular cut-off, and the subgroup of patient with RVEF 45% is by far too small to be relevant by itself or comparing to the RVEF<45% group.

- Thank you for your important indication. We have clarified in the text that we chose the RVEF <45% cut-off value in agreement with prior publications studying RV dysfunction by CMR, two of which are cited in references 17 and 18:

Ref 17. Petersen SE, Aung N, Sanghvi MM, Zemrak F, Fung K, Paiva JM, Francis JM, Khanji MY, Lukaschuk E, Lee AM, et al. Reference ranges for cardiac structure and function using cardiovascular magnetic resonance (CMR) in Caucasians from the UK Biobank population cohort. J Cardiovasc Magn Reson. 2017;19:18. doi: 10.1186/s12968-017-0327-9.

Ref 18. Aschauer S, Kammerlander AA, Zotter-Tufaro C, Ristl R, Pfaffenberger S, Bachmann A, Duca F, Marzluf BA, Bonderman D, Mascherbauer J. The right heart in heart failure with preserved ejection fraction: insights from cardiac magnetic resonance imaging and invasive haemodynamics. Eur J Heart Fail. 2016;18:71-80. doi: 10.1002/ejhf.418

In addition, there are many other examples studies that have also employed the cut-off value of RVEF <45% to define RV dysfunction by CMR:

- Gulati A, Ismail TF, Jabbour A, Alpendurada F, Guha K, Ismail NA, Raza S, Khwaja J, Brown TD, Morarji K, Liodakis E, Roughton M, Wage R, Pakrashi TC, Sharma R, Carpenter JP, Cook SA, Cowie MR, Assomull RG, Pennell DJ, Prasad SK. The prevalence and prognostic significance of right ventricular systolic dysfunction in nonischemic dilated cardiomyopathy. Circulation. 2013;128:1623-33. doi: 10.1161/CIRCULATIONAHA.113.002518.

- Focardi M, Cameli M, Carbone SF, Massoni A, De Vito R, Lisi M, Mondillo S. Traditional and innovative echocardiographic parameters for the analysis of right ventricular performance in comparison with cardiac magnetic resonance. Eur Heart J Cardiovasc Imaging. 2015;16:47-52. doi: 10.1093/ehjci/jeu156.

- Jolly, U.S., Nevis, I., Almehmadi, F.S. et al. Influence of right ventricular ejection fraction on the occurrence of arrhythmic events in patients with systolic dysfunction. J Cardiovasc Magn Reson 16, O32 (2014). https://doi.org/10.1186/1532-429X-16-S1-O32.

Albeit the number of patients with RVEF <45% in our sample was limited (20% of the sample), we believe the fact that low mvPA values still identified patients with worse outcomes among patients with preserved RV function, a larger group of 46 individuals is relevant. This way, lower mvPA values act as early prognostic indicators, even in the absence of RV dysfunction which would indicate a more advanced state of the disease.

> Moreover, looking at the ROC curve, sensibility and specificity don’t look awesome. Can the authors specify them and discuss the risk of misdiagnosis and misclassification?

- The sensitivity of the cut-off value of 9cm/s of mvPA is 81%, while its specificity is 65%. In the context of identifying HF patients at higher risk of decompensation and/or death, we believe it is more important that our test-variable (mvPA) has a greater sensitivity than specificity, as this variable should aim at identifying the largest possible number of “high-risk” individuals. On the other hand, while the relatively low specificity of this cut-off value (mvPA 9cm/s) is not ideal, it will not derive into any “unnecessary” additional therapies and/or further (risky) examinations, which are currently not indicated in this group of patients, whichever their risk of events might be, but rather, into a closer ambulatory surveillance of the patient.

> As left atrial volume is a predictor of the primary endpoint, can the authors analyze the ROC curve of left atrial volume and show mvPA is effectively a better predictor than the widely used left atrial volume.

- We have analyzed the ROC curve of left atrial (LA) area determined by CMR and have now included it in Figure 3a. The optimal cut-off value for LA area was 19cm2, with an AUC of 0.74 (95% confidence interval 0.59-0.90, p-value: 0.004). Albeit this AUC value is very similar to that of mvPA, we believe mvPA is more useful in the clinical context of identifying HF patients at a higher risk of adverse events as its cutoff value has a higher sensitivity. This way, the risk of not being able to identify a “high-risk” patient as such is much lower for mvPA than it is for LA area. While the consequences of “over-diagnosing” or labelling as “high-risk” a patient that is not at such high risk are not serious in this context, as no invasive additional therapies will be implemented, but rather, a closer and more thorough clinical surveillance.

In addition, an enlarged left atrium not only reflects a status of chronic congestion but can also be caused by other conditions such as atrial fibrillation (AF), which could reduce the accuracy of this parameter as a tool to identify higher-risk patients. Indeed, almost half of the patients in our sample presented AF, so that left atrial area´s cut-off value that best predicted outcomes in our sample would probably not apply to other populations that had a lower prevalence of  AF.

> Can the authors precise the indicators included in their univariate and multivariate analysis mentioned line 246-248. Why mvPA was not included as a continuous variable in this analysis?

- Thank you for your interesting considerations. Every parameter with known or suspected prognostic value was included on univariate analysis, as listed below these lines:

  • Age, sex, diabetes mellitus, arterial hypertension, coronary or cerebrovascular disease, NYHA functional class, prior HF hospitalization, kidney function, atrial fibrillation, left bundle branch block, NT-proBNP.
  • Left ventricular ejection fraction (LVEF) and mass, indexed left atrial volume, diastolic pattern parameters (E/A ratio, E/e´ratio..), tricuspid annular plane systolic excursion (TAPSE), systolic pulmonary artery pressure (SPAP), TAPSE/SPAP ratio,
  • LVEF(%), RVEF(%), end-diastolic and end-systolic ventricular volumes, left atrial and right atrial areas, late gadolinium enhancement, mvPA, maximal and minimal PA area, PA pulsatility, pulmonary vascular resistances, pulmonary artery pressures (mean PA pressure, PA wedge pressure, PA pulse pressure...), cardiac output, PA compliance.

- As to multivariate analysis, as specified in the manuscript, we included every parameter with a p-value < 0.10 on univariate analysis. To avoid colinearity, those variables that reflected the same clinical parameter, e.g.: mvPA and PA maximal area, both reflecting indirectly the distensibility of the pulmonary circulation, we evaluated them against each other on Cox multivariable regression analysis and discarded variables that did not maintain statistical significance from the final model. The indicators included in the final model are displayed in table 4:

Table 4. Multivariate Regression analysis.

Hazard Ratio (95% CI)

p-value

NYHA functional class III-IV/IV

0.67 (0.24-1.90)

0.453

Left atrial area - CMR (ml/m2)

1.12 (1.01-1.24)

0.034

Late gadolinium enhancement – CMR

0.72 (0.24-2.12)

0.546

mvPA<9cm/s

4.11(1.28-13.19)

0.017

CMR: cardiac magnetic resonance. NYHA: New York heart association. mvPA: mean velocity pulmonary area

- In addition, we included mvPA as a dichotomous variable in Cox regression analysis, as we believe it is easier for clinicians to interpret and apply into clinical practice a new parameter if they have a reference value against which they can contrast their patients results. By providing the 9cm/s cut-off value of mvPA, it becomes easier for clinicians to understand where normal and abnormal values stand. Notwithstanding, mvPA also presented independent prognostic value on Cox regression when included as a continuous variable:

Multivariate Regression analysis.

Hazard Ratio (95% CI)

p-value

NYHA functional class III-IV/IV

0.69 (0.24-1.96)

0.489

Left atrial area - CMR (ml/m2)

1.12 (1.01-1.25)

0.031

Late gadolinium enhancement – CMR

0.86 (0.30-2.44)

0.777

mvPA (cm/s)

1.21 (1.02-1.45)

0.036

CMR: cardiac magnetic resonance. NYHA: New York heart association. mvPA: mean velocity pulmonary area

> Right Heart Catheterization was performed in a subset of patients. As it is a retrospective study and this test was performed on the only basis of physicians’ judgment, there is unavoidable bias. Demographic, CMR and echocardiographic data should be displayed (at least as a part of the supplemental data) to compare patients in whom RHC was done and those in whom it wasn’t.

- Thank you for your interesting comment. Indeed, RHC was performed in a relatively low number of patients in our sample and there is an unavoidable bias regarding individuals who underwent or not RHC. We have now included a supplementary table (supplementary Table 2) with patients´ baseline and imaging characteristics according to whether or not they underwent RHC. Patients undergoing RHC more frequently displayed a more advanced functional class, RV dysfunction and elevated estimated pulmonary pressures by echocardiography. On CMR, patients undergoing RHC displayed a greater prevalence of LGE, lower mvPA values and higher estimated PVR. To sum up, patients that underwent a RHC were frequently “sicker” than those in which the physician did not indicate this examination.

> In the “limitations” section, authors stated about the loss of follow-up for some patients. Are they writing about the nine patients mentioned line 73 or are there other losses?

- Thank you for your important observation. We are referring to the 9 patients mentioned in line 73.

> In the introduction (line 55 and subsequent), it would be more accurate to state the authors’ team studied mvPA as a potential measurement of interest on monocentric small cohorts rather than presenting it as an established measurement ; or  please provide other references from other teams.

- Thank you for your comment. We have reworded the introduction to stress the fact that, to the best of our knowledge, mvPA has not been studied by other teams other than ours to date.

Reviewer 2 Report

The research team has experience on this CMR-Marker and the main publications have been written by them. HFPEF is a more detailed disease model than that with reduced ejection fraction.

Some personal observations
- in the introduction you explain that the pressure-volume curves that are studied in books derive from experimental models, in vivo it is complex and surrogate markers are used
- How do you explain that lower mvPA values ​​are prognostic? What matters most? Right ventricular dysfunction (and low flow)?
- Are lower velocities associated with higher pressures? It is counterintuitive to echocardiography measures. 
- I would like to see, if possible, some correlation curves between mvPA and right catheterization, in particular the Wedge pressure. Alternatively, an exemplary clinical case with resonance and catheterization (if possible)
- Right ventricular function is very important in HFPEF. However, in the acute event (re-hospitalization for decompensation) the ventricular function worsens and the wedge increases (obviously only in some HFPEF phenotypes). Do you think this marker may be suitable for a somewhat intermittent disease?
- Put a CMR measurement figure of mvPA with peak velocity and area.
- are there any BNP values?
- the patient with pulmonary hypertension is a "sui generis" patient with unpredictable clinical responses (DOI: 10.1177 / 2045894020956581) and increasing attention must be paid to the pulmonary artery diameter by all cardiovascular imaging methods (doi: 10.1007 / s00330-020-07622-x. Epub 2020 Dec 23). See if you can add at least one of these references. 
- Even with the limits of my knowledge, I can only recognize the merits of your work. Feel free to judge some of my comments, it's a new marker and I'm much more familiar with right catheterization and ultrasound.

Author Response

Response to reviewers.

REVIEWER 2:

Comments and Suggestions for Authors

The research team has experience on this CMR-Marker and the main publications have been written by them. HFPEF is a more detailed disease model than that with reduced ejection fraction.

Some personal observations

> In the introduction you explain that the pressure-volume curves that are studied in books derive from experimental models, in vivo it is complex and surrogate markers are used.

-Thank you for your interesting comment. Indeed, pressure-volume analysis is the gold standard metric to assess ventricular performance. Albeit this technique has been employed in research, including humans, its use in routine clinical practice is not indicated as it requires invasive pressure measurements by means of right heart catheterization as well as specific dedicated material, i.e., specific conductance catheters.

> How do you explain that lower mvPA values are prognostic? What matters most? Right ventricular dysfunction (and low flow)?

-Thank you for your important observation. We believe that low PA velocities are prognostic as they simultaneously reflect two important factors, both of which a prognostically significant: right ventricular (RV) output and compliance of the pulmonary circulation. On the one hand mvPA reflects RV function and stroke volume: in the absence of RV outflow tract obstruction or pulmonary valve dysfunction, the volume of blood ejected into the PA will be smaller and acquire a lower speed in patients with RV dysfunction as opposed to those with normal RV function. On the other hand, decreased PA compliance will lead to an increase in pulsatile RV afterload as it generates a premature reflection of blood-flow waves from the distal vessels to the proximal conduit arteries in late systole, thus decreasing the net value of mvPA.

> Are lower velocities associated with higher pressures? It is counterintuitive to echocardiography measures.

- Thank you for your comment. In our sample, lower mvPA were associated with reduced right ventricular ejection fraction on CMR but not with higher pulmonary artery (PA) and pulmonary capillary wedge pressures on RHC nor systolic PA pressure estimated by TTE.

Similarly, no association between mvPA and systolic PA pressure by TTE nor

mean PA pressure and pulmonary capillary wedge pressures on RHC was observed in the prior study by our group that included patients with HFrEF (doi: 10.1186/s12968-020-00621-3). In the latter study, which included a larger number of patients undergoing RHC, lower mvPA was associated with greater PA pulse pressure, transpulmonary gradients and lower PA compliance on RHC.

> I would like to see, if possible, some correlation curves between mvPA and right catheterization, in particular the Wedge pressure. Alternatively, an exemplary clinical case with resonance and catheterization (if possible).

- Thank you for your comment. In our sample, we found no correlation between mvPA and RHC parameters, including wedge pressure, probably in relation to the low number of patients undergoing RHC. We have provided an exemplary clinical case in the new Figure 1 comprising CMR and RHC values to show the relationship between the values obtained by each of these techniques.

> Right ventricular function is very important in HFPEF. However, in the acute event (re-hospitalization for decompensation) the ventricular function worsens and the wedge increases (obviously only in some HFPEF phenotypes). Do you think this marker may be suitable for a somewhat intermittent disease?

- Thank you for your important consideration. Indeed, previous studies have found that patients with elevated pulmonary artery (PA) and left atrial (LA) pressures are at higher risk of subsequent heart failure (HF) hospitalizations. Although triggers for HF decompensations are somewhat unpredictable, they are more likely to lead to events when starting from a higher plateau, or if the pulmonary circulation (PC) is non-compliant and thus, less able to accommodate an excess volume overload.

Ambulatory monitoring of PA and LA pressures requires placement of intracardiac monitoring devices, which limits its widespread adoption in clinical practice. Therefore, the majority of studies that stratify prognosis in HF have focused on non-invasive imaging-derived indicators of elevated filling pressures.

We believe that mvPA, as a marker of right ventricular (RV) to PC coupling, can be useful to identify patients at a higher risk of HF hospitalizations as it is closely linked with LA pressures and also reflects the ability of the PC to adapt to volume challenges. Indeed, several studies have found that elevated PA capillary wedge pressure increases the net RV afterload, by elevating the pulsatile load exerted upon the RV, which has a direct negative effect on RV to PC coupling and decreases PA compliance at any given value of pulmonary vascular resistances (1). Impairment in LA reservoir function determined by echocardiography has also been linked with decreased PA compliance and RV to PC uncoupling, both at rest and during exercise (2-3). For all the above, we believe mvPA is a valuable variable to predict HF events.

  • Tedford RJ, Hassoun PM, Mathai SC, Girgis RE, Russell SD, Thiemann DR, Cingolani OH, Mudd JO, Borlaug BA, Redfield MM, Lederer DJ, Kass DA. Pulmonary capillary wedge pressure augments right ventricular pulsatile loading. Circulation. 2012;125:289-97.
  • Sugimoto T, Bandera F, Generati G, Alfonzetti E, Bussadori C, Guazzi M. Left Atrial Function Dynamics During Exercise in Heart Failure: Pathophysiological Implications on the Right Heart and Exercise Ventilation Inefficiency. JACC Cardiovasc Imaging. 2017;10:1253-64.
  • Melenovsky V, Hwang SJ, Redfield MM, Zakeri R, Lin G, Borlaug BA. Left atrial remodeling and function in advanced heart failure with preserved or reduced ejection fraction. Circ Heart Fail. 2015;8:295-303.

> Put a CMR measurement figure of mvPA with peak velocity and area.

- Thank you for your important indication. We have now included an additional figure (new Figure 1) that displays an example of a CMR measurement of PA area as well as mvPA and peak PA velocity in a velocity-encoded gradient echo sequence.

> are there any BNP values?

- Thank you for your important comment. NT-proBNP values are included in Table 1 and Supplementary Table 1 depicting baseline clinical and analytical characteristics of patients according to the primary combined endpoint and mvPA values, respectively.

> The patient with pulmonary hypertension is a "sui generis" patient with unpredictable clinical responses (DOI: 10.1177 / 2045894020956581) and increasing attention must be paid to the pulmonary artery diameter by all cardiovascular imaging methods (doi: 10.1007 / s00330-020-07622-x. Epub 2020 Dec 23- Chest CT–derived pulmonary artery enlargement at the admission predicts overall survival in COVID-19 patients: insight from 1461 consecutive patients in Italy). See if you can add at least one of these references.

- Thank you for interesting comment. Indeed, enlarged main pulmonary artery diameters measured either by CT or CMR have been associated with higher pulmonary artery pressures and a worse prognosis in a variety of conditions such as HF, COPD and recently COVID-19. We have now included your reference in the manuscript.

> Even with the limits of my knowledge, I can only recognize the merits of your work. Feel free to judge some of my comments, it's a new marker and I'm much more familiar with right catheterization and ultrasound.

- Thank you for comment.

Round 2

Reviewer 1 Report

I carefully authors' responses to the comments I made. Responses are acurate and bring valuable precisions to the questions I raised. The manuscript is more balanced and I have no more comment to make